# *WAPO-A1* is the causal gene of the 7AL QTL for spikelet number per spike in wheat

Saarah Kuzay[1‡], Huiqiong Lin[1,2‡], Chengxia Li[1,2], Shisheng Chen[1,3], Daniel P. Woods[1,2], Junli Zhang[1], Tianyu Lan[4], Maria von Korff[4,5], Jorge Dubcovsky[1,2]*

**1** Department of Plant Sciences, University of California, Davis, California, United States of America, **2** Howard Hughes Medical Institute, Chevy Chase, Maryland, United States of America, **3** Peking University Institute of Advanced Agricultural Sciences, Weifang, Shandong, China, **4** Institute for Plant Genetics, Heinrich Heine University, Duesseldorf, Germany, **5** Cluster of Excellence on Plant Sciences "SMART Plants for Tomorrow's Needs", Heinrich Heine University, Duesseldorf, Germany

‡ SK and HL contributed equally to this work.
* jdubcovsky@ucdavis.edu

**Data Availability Statement:** We deposited the Kronos near isogenic line with the H2 introgression in the National Small Grains Collection (PI 698810). All data are presented in the text and supplementary materials. The raw data for all

## Abstract

Improving our understanding of the genes regulating grain yield can contribute to the development of more productive wheat varieties. Previously, a highly significant QTL affecting spikelet number per spike (SNS), grain number per spike (GNS) and grain yield was detected on chromosome arm 7AL in multiple genome-wide association studies. Using a high-resolution genetic map, we established that the A-genome homeolog of *WHEAT ORTHOLOG OF APO1* (*WAPO-A1*) was a leading candidate gene for this QTL. Using mutants and transgenic plants, we demonstrate in this study that *WAPO-A1* is the causal gene underpinning this QTL. Loss-of-function mutants *wapo-A1* and *wapo-B1* showed reduced SNS in tetraploid wheat, and the effect was exacerbated in *wapo1* combining both mutations. By contrast, spikes of transgenic wheat plants carrying extra copies of *WAPO-A1* driven by its native promoter had higher SNS, a more compact spike apical region and a smaller terminal spikelet than the wild type. Taken together, these results indicate that *WAPO1* affects SNS by regulating the timing of terminal spikelet formation. Both transgenic and *wapo1* mutant plants showed a wide range of floral abnormalities, indicating additional roles of *WAPO1* on wheat floral development. Previously, we found three widespread haplotypes in the QTL region (H1, H2 and H3), each associated with particular *WAPO-A1* alleles. Results from this and our previous study show that the *WAPO-A1* allele in the H1 haplotype (115-bp deletion in the promoter) is expressed at significantly lower levels in the developing spikes than the alleles in the H2 and H3 haplotypes, resulting in reduced SNS. Field experiments also showed that the H2 haplotype is associated with the strongest effects in increasing SNS and GNS (H2>H3>H1). The H2 haplotype is already present in most modern common wheat varieties but is rare in durum wheat, where it might be particularly useful to improve grain yield.

figures and Supplemental Tables are available in S1 Data file.

**Funding:** JD received support for this project from the Agriculture and Food Research Initiative Competitive Grants 2017-67007-25939 (WheatCAP), USDA National Institute of Food and Agriculture (NIFA, https://nifa.usda.gov/) and from the Howard Hughes Medical Institute (https://www.hhmi.org/). The USDA-NIFA grant supported the salaries of SK, JZ and SC. The Howard Hughes Medical Institute supported the salaries of JD, HL, CL and DW. DW is a Howard Hughes Medical Institute Fellow of the Life Sciences Research Foundation (http://www.lsrf.org/) that paid his salary for three years. The funders had no role in study design, data collection and analysis, decision to publish, or preparation of the manuscript.

**Competing interests:** The authors have declared that no competing interests exist.

## Author summary

A region on wheat chromosome 7A has been previously shown to affect the number of spikelets and grains per spike as well as total grain yield in multiple breeding programs. In this study, we show that loss-of-function mutations in the *WAPO1* gene located within this region reduce the number of spikelets per spike and that additional transgenic copies of this gene increase this number. These results demonstrate that *WAPO1* is the gene responsible for the differences in grain number and yield associated with the 7A chromosome region. Among the three main variants identified for this gene, we demonstrate in field experiments that the H2 variant is associated with the largest increases in number of spikelets and grains per spike. The H2 *WAPO1* variant is frequent in bread wheat breeding programs but is almost absent in modern pasta wheat varieties. Therefore, the introgression of the H2 represents a promising opportunity to improve grain yield in pasta wheat.

## Introduction

Wheat is an essential staple crop for global food security. It is highly adapted to a wide variety of climates and production systems, and provides more than 20% of the calories and protein consumed by the human population [1]. Although further increases in grain yield are required to feed a continuously growing population, historical yield trend studies have shown a decrease in the relative rates of grain yield gains in some wheat growing regions [2]. This has prompted new efforts to understand and improve the productivity of both common (*Triticum aestivum*, genomes AABBDD) and durum wheat (*T. turgidum* ssp. *durum*, genomes AABB).

Identifying genes controlling total grain yield is challenging due to its complex quantitative nature and genotype by environment interactions [3]. However, grain yield can be dissected into more discrete yield components with higher heritability. Total grain yield can be partitioned into several yield components, including number of spikes per area unit, spikelet number per spike (SNS), grain number per spikelet, and average grain weight. Among these traits, SNS usually exhibits high heritability because it is established early in the reproductive phase when the terminal spikelet is formed [4], limiting the effect of environmental conditions after this point.

A highly significant and stable QTL for SNS was identified on chromosome arm 7AL in multiple genome-wide association studies (GWAS) including a panel of soft red winter wheats in the US [5], panels of European winter wheats [6–9], a panel of US and CIMMYT photoperiod-insensitive spring wheats, and six biparental populations which comprised different wheat market classes [10]. In our previous study, we generated two high-resolution genetic maps to delimit this SNS QTL to an 87-kb region (674,019,191–674,106,327 bp, RefSeq v1.1) containing four candidate genes [10]. Among these genes, we identified *TraesCS7A02G481600* as the most promising candidate gene, based on the presence of a non-synonymous polymorphism that co-segregated with SNS in biparental populations segregating for different haplotypes in the candidate region [10].

The wheat gene *TraesCS7A02G481600* is orthologous to the *Oryza sativa* (rice) gene *ABERRANT PANICLE ORGANIZATION1* (*APO1*), hence it was designated as *WHEAT ORTHOLOG of APO1* (*WAPO1*). Loss-of-function mutants in rice *APO1* reduce panicle branching and spikelet number [11], supporting *WAPO-A1* as a promising candidate gene for the SNS QTL [7,8,10]. The rice *APO1* gene and its homolog in *Arabidopsis thaliana* (Arabidopsis), *UNUSUAL FLORAL ORGANS* (*UFO*), encode an F-box protein that is a component of an SCF

(**S**kp1–**C**ullin–**F**-box-protein) ubiquitin ligase [12,13]. This domain is important to maintain the activity of LEAFY (LFY), a transcription factor that plays key roles in flowering and floral development [14].

In rice, mutations in *APO1* or *LFY* (also known as *APO2* and *RFL* in rice) result in reductions in the number of branches and spikelets per panicle. The effect is similar in the *apo1 lfy* double mutant, suggesting that these two genes act cooperatively to control this trait [15]. Mutations in these two genes are also associated with floral abnormalities, with more severe phenotypes in the *apo1 lfy* double mutant than in either of the two single mutants. These results suggest that these two genes also play important roles in floral development [15]. Floral defects in the rice *apo1* and Arabidopsis *ufo* mutants are concentrated to the internal floral whorls [12,13].

Rapid changes in *WAPO-A1* allele frequencies during wheat domestication and breeding suggest that this region is relevant to wheat improvement [10]. Three major haplotypes were identified in the 87-kb candidate gene region–H1, H2, and H3 –each of which associated with different *WAPO-A1* alleles. Haplotype H3 includes the ancestral alleles *Wapo-A1c* and *Wapo-A1d*, which differ from each other by two synonymous substitutions, two SNPs in the single intron and one in the promoter, which likely have limited effect on gene function [10]. Haplotype H3 is present in the diploid donor of the A genome (*T. urartu*), cultivated emmer (*T. turgidum* ssp. *dicoccon*) and wild emmer (*T. turgidum* ssp. *dicoccoides*), and at low frequency in modern durum and common wheat varieties. Haplotype H1, present in over 99% of modern durum wheat varieties, has the *Wapo-A1a* allele that is characterized by a 115-bp deletion in the promoter and a change from aspartic acid to asparagine at position 384 (D384N). This amino acid change is predicted to have a limited effect on protein structure and function (Table 1, BLOSUM62 score = 1). *WAPO-A1* haplotype H2, the most frequent haplotype in modern common wheat varieties, carries the *Wapo-A1b* allele and differs from the ancestral haplotype by a cysteine to a phenylalanine polymorphism at amino acid position 47 (C47F) in a conserved region of the F-box motif [7,8,10]. This amino acid change is predicted to have a strong effect on protein structure and function (Table 1, BLOSUM62 score = -2). Linkage analysis in six different biparental populations established that the H2 haplotype was associated with higher SNS than both the H1 and H3 haplotypes [10].

Our previous study established *WAPO-A1* as the best candidate gene for the 7AL SNS QTL [10], but functional validation was missing. In this study, we demonstrate that *WAPO1* is the

**Table 1. Comparison of regulatory and coding *WAPO-A1* regions in wild type Kronos (H1 haplotype) and the three genomic constructs used in the transgenic plants.**

| *WAPO-A1* haplotype | | | | H1 | TmDV92 | H3 | H2 |
|---|---|---|---|---|---|---|---|
| RefSeq v1.1 | DNA change | Protein effect | BLOSUM 62 score [a] | Kronos | TmDV92 | LDN-C47 | LDN-F47 [b] |
| **Promoter** | | | | | | | |
| 674,080,862 | 115 bp del | none | n/a | **present** | absent | absent | absent |
| **Coding Region** | | | | | | | |
| 674,081,601 | G140T | C47F | -2 | C | C | C | **F** |
| 674,081,843 | C382G | P128A | -1 | P | **A** | P | P |
| 674,082,413 | A952G | Q273R | 1 | Q | **R** | Q | Q |
| 674,082,673 674,082,674 | G1212A C1213T | A360M | -1 | A | **M** | A | A |
| 674,082,745 | A1284G | N384D | 1 | **N** | D | D | D |

[a] The more negative the BLOSUM 62 score is, the most likely is the change to affect protein structure or function.

[b] This construct includes the edited F47 polymorphism in LDN-C47, but none of the promoter SNPs present in the natural haplotype H2 [10].

gene underpinning the SNS QTL by characterizing loss-of-function mutants and transgenic plants and exploring its spatial and temporal distribution in the developing spike. We also describe the flower abnormalities observed in plants with complete loss of *WAPO1* activity and in transgenic plants with additional *WAPO1* genes. Finally, we characterize the effect of different natural *WAPO-A1* alleles on the number of spikelets and grains per spike and discuss their potential applications in common and durum wheat breeding programs.

## Results

### Loss-of-function EMS and CRISPR mutations in *WAPO1* reduce spikelet number per spike

A search of the sequenced EMS-mutagenized population of tetraploid wheat Kronos [16], which carries the *WAPO-A1* H1 haplotype, yielded one mutant line with a premature stop codon in *WAPO-A1*, but no stop codon or splicing site mutations were detected in the *WAPO-B1* homeolog. The *WAPO-A1* mutation W216* identified in Kronos mutant line K4222 results in a premature stop codon and the truncation of 51% of the encoded protein, which has 440 amino acids. The truncated region includes sequences highly conserved among grasses, suggesting that the truncated protein is either non-functional or has very reduced activity. The modified activity of the W216* truncated gene was confirmed by the significant SNS reduction in lines carrying the *wapo-A1* mutant allele relative to those carrying the wild type (WT) allele in three different experiments (Fig 1). In the greenhouse experiment, the average SNS of $F_3$ lines homozygous for *wapo-A1* EMS mutant was 1.7 spikelets lower (14% reduction, $P < 0.001$) than the sister lines homozygous for the WT allele (Fig 1A). A similar reduction of 1.4 spikelets (7.3% reduction, $P < 0.001$) was observed in a field experiment using homozygous $F_4$ sister lines for the *wapo-A1* and WT alleles (Fig 1B). The number of florets per spikelet was not affected in the mutants.

To reduce potential variability originated from background mutations and more accurately determine the effect of the W216* truncation, we crossed the K4222 mutant to WT Kronos twice and selected homozygous $BC_1F_2$ sister lines for an additional field evaluation using their $BC_1F_3$ grains. The $BC_1F_3$ plants homozygous for *wapo-A1* had an average of 1.1 fewer spikelets per spike relative to the WT sister line (6.0% reduction, $P = 0.0035$, Fig 1C). Taken together, these three experiments indicate that loss-of-function of the A-genome homeolog of *WAPO1* is sufficient to significantly reduce SNS.

To test the effect of the *WAPO-B1* homeolog and validate the results for the EMS induced *wapo-A1* mutant, we generated three independent $T_0$ CRISPR-Cas9 transgenic Kronos plants with a guide RNA targeting both homeologs (S1 Table). We identified one line with a "T" insertion at position 510 from the ATG (4-bp upstream of the CGG PAM site) in both *WAPO-A1* and *WAPO-B1*, and selected it for further characterization. This frameshift insertion alters 61.4% of the protein sequence (starting from amino acid 171), likely resulting in loss-of-function of both *WAPO1* homeologs. We genotyped 110 $T_1$ plants derived from the selected transgenic event using the *WAPO-A1* and *WAPO-B1* CAPS markers described in S1 Table. The proportion of lines homozygous for the WT *Wapo-A1* (2.7%) and/or *Wapo-B1* (4.5%) alleles was significantly lower than the expected 25%, suggesting continuous CRISPR editing. This hypothesis was validated by the identification of a novel 5-bp frame-shift deletion starting at position 505 in *WAPO-B1* (4-bp upstream of the PAM site) in line $T_1$-40, which was re-sequenced and crossed to WT Kronos to eliminate the CRISPR-Cas9 transgene.

We created lines homozygous for all four possible combinations of the *WAPO1* homeologs–WT, *wapo-A1*, *wapo-B1*, and *wapo1* double mutant–by selecting $F_2$ progeny derived from an $F_1$ plant without the CRISPR-Cas9 transgene. In a growth chamber experiment, we

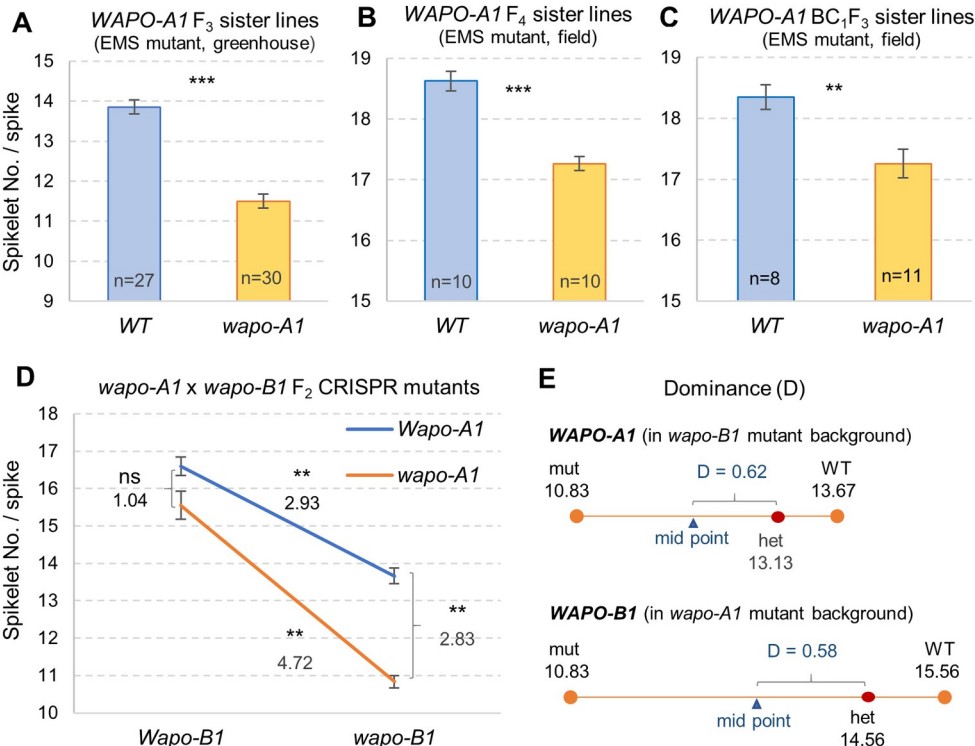

**Fig 1. Differences in spikelet number per spike (SNS) in *WAPO1* mutant lines.** (**A-C**) Homozygous Kronos sister lines segregating for the EMS induced *wapo-A1* W216* mutation. (**A**) $F_3$ sister lines evaluated in the greenhouse. (**B**) $F_4$ sister lines evaluated in the field. (**C**) $BC_1F_3$ sister lines evaluated in the field. (**D**) Loss-of-function CRISPR induced *wapo-A1* and *wapo-B1* mutations. Interaction graph showing the effect of *wapo-A1* and *wapo-B1* in tetraploid wheat Kronos plants without the CRISPR-Cas9 transgene. (**E**) Degree of dominance [17] for *WAPO-A1* (in *wapo-B1* background) and *WAPO-B1* (in *wapo-A1* background). Bars are average SNS and error bars are s.e.m. ns = not significant, * = $P < 0.05$, ** = $P < 0.01$ and *** = $P < 0.001$. Data and statistical analyses used for this figure are available in Data A in S1 Data and panel A in S2 Table.

detected highly significant differences for SNS (panel A in S2 Table), but non-significant differences for heading date (panel B in S2 Table) or leaf number (panel C in S2 Table). Both *wapo-A1* and *wapo-B1* mutants showed significantly lower SNS than the WT (Fig 1D, panel A in S2 Table and Data A in S1 Data), with larger reductions in *wapo-B1* (3 spikelets) than in *wapo-A1* (1 spikelet). The transcript levels of the two homeologs were also different, with higher levels of *WAPO-B1* than *WAPO-A1* at different stages of spike development (S1 Fig). The effect of the *wapo-A1* allele on SNS was highest when the B-genome homeolog was homozygous for *wapo-B1*, and vice versa, resulting in a highly significant interaction ($P = 0.0096$, Fig 1D, and panel A in S2 Table). Partial dominance at the *WAPO-A1* (D = 0.61) and *WAPO-B1* (D = 0.58) loci (Fig 1E) indicates partial functional redundancy between the two copies within each of the *WAPO1* homeologs.

## Transgenic plants with additional *WAPO1* copies show delayed heading time and higher SNS

To compare the effect of three *WAPO-A1* alleles encoding proteins with different non-synonymous polymorphisms (Table 1), we transformed three different constructs into Kronos. We used the bacterial artificial chromosome (BAC) libraries of diploid *T. monococcum* (accession DV92) [18] and tetraploid wheat Langdon (LDN-C47) [19] to clone the genomic regions of

*WAPO-A1* from these two species. The cloned regions included the intron and regulatory regions (see Materials and Methods). We then used site-directed mutagenesis to change the cysteine at position 47 in LDN-C47 (H3 haplotype) into a phenylalanine (LDN-F47), resulting in a protein identical to the one encoded by the *Wapo-A1b* allele in the H2 haplotype (Table 1).

We transformed the three constructs into Kronos and obtained four independent transgenic events for TmDV92, three for LDN-C47, and five for the LDN-F47 derived allele, and evaluated their corresponding $T_1$ progeny in greenhouse experiments for SNS (Fig 2). Transgenic plants for all three constructs showed similar increases in SNS relative to the WT: LDN-C47 (13.4%), TmDV92 (15.1%) and LDN-F47 (15.5%) (Fig 2). Increases in SNS were significant for two TmDV92 events, one LDN-C47 event (Fig 2A) and four LDN-F47 independent events relative to the WT control (Fig 2C).

The effects of the transgenes on heading date were smaller and slightly less significant than those for SNS. For heading date, a 7.0% delay was observed in transgenic plants with the LDN-C47 construct, a 5.4% delay for TmDV92, and a 6.3% delay for LDN-F47 (Fig 2B and 2D). Heading date was positively and significantly correlated with SNS among the $T_1$ lines segregating for each of the three constructs ($R = 0.53–0.87$, $P < 0.001$, Data B in S1 Data). Taken together, these results demonstrate that the presence of additional copies of *WAPO1* in the transgenic plants increases SNS and delays heading date.

## CRISPR *wapo1* mutant shows altered floral morphology

In addition to the reduced SNS, *wapo1* double mutant displayed a great diversity of floral abnormalities and produced a limited number of grains. To establish the frequency of the different floral abnormalities, we dissected 91 florets from 32 spikelets located at different positions along the spike from 14 different *wapo1* mutant plants (Data C in S1 Data). While glumes, lemmas and paleae were normal, multiple abnormalities were observed in other floral organs. The number of lodicules, which is normally two per floret, varied from zero to four (average 1.95, Fig 3A). Moreover, lodicules were frequently fused with stamens (19.1%), leaf-like tissue (22.5%) or both (5.6%, Fig 3B–3D). The number of stamens per floret was lower than three in 42% of the florets (Fig 3A–3D), resulting in an average of 2.15 stamens per floret. Stamens were frequently fused with each other or with ovaries (6.6%), lodicules (18.9%), leaf-like tissue (9.2%) or combinations of two of the previous three categories (4.6%, Fig 3B and 3C). Most florets had one pistil with one ovary, but 43% of the florets had multiple pistils, likely due to loss of floret meristem determinacy. In 28.4% of the florets, the ovaries were fused with leafy tissue (Fig 3D and Data C in S1 Data). For comparison, a wild-type Kronos floret is shown in Fig 4.

Since most of the floral abnormalities in *wapo1* were detected in the internal whorls (lodicules, stamens and pistils), we explored the effect of the *wapo1* mutant on the expression of class-B, -C and -E MADS-box floral genes in developing spikes at the stamen primordia stage. Relative to the WT, the *wapo1* mutant showed a significant down regulation of class-B (*AP3-1* and *PI1*) and class-C (*AG1* and *AG2*) MADS-box genes (Fig 3E and 3F), but no significant differences were detected for the control *SEPALLATA* MADS-box genes *SEP1*-2, *SEP1*-4, *SEP1*-6, or *SEP3-2* (Fig 3G).

## *WAPO1* transgenic plants exhibit abnormal spike and floral phenotypes

Among the $T_1$ plants segregating for the three transgenic *WAPO1* genes driven by their native promoters, the events showing significant differences in SNS also displayed spike abnormalities (Fig 4). One unusual phenotype was the presence of naked pistils at the base of spikelets

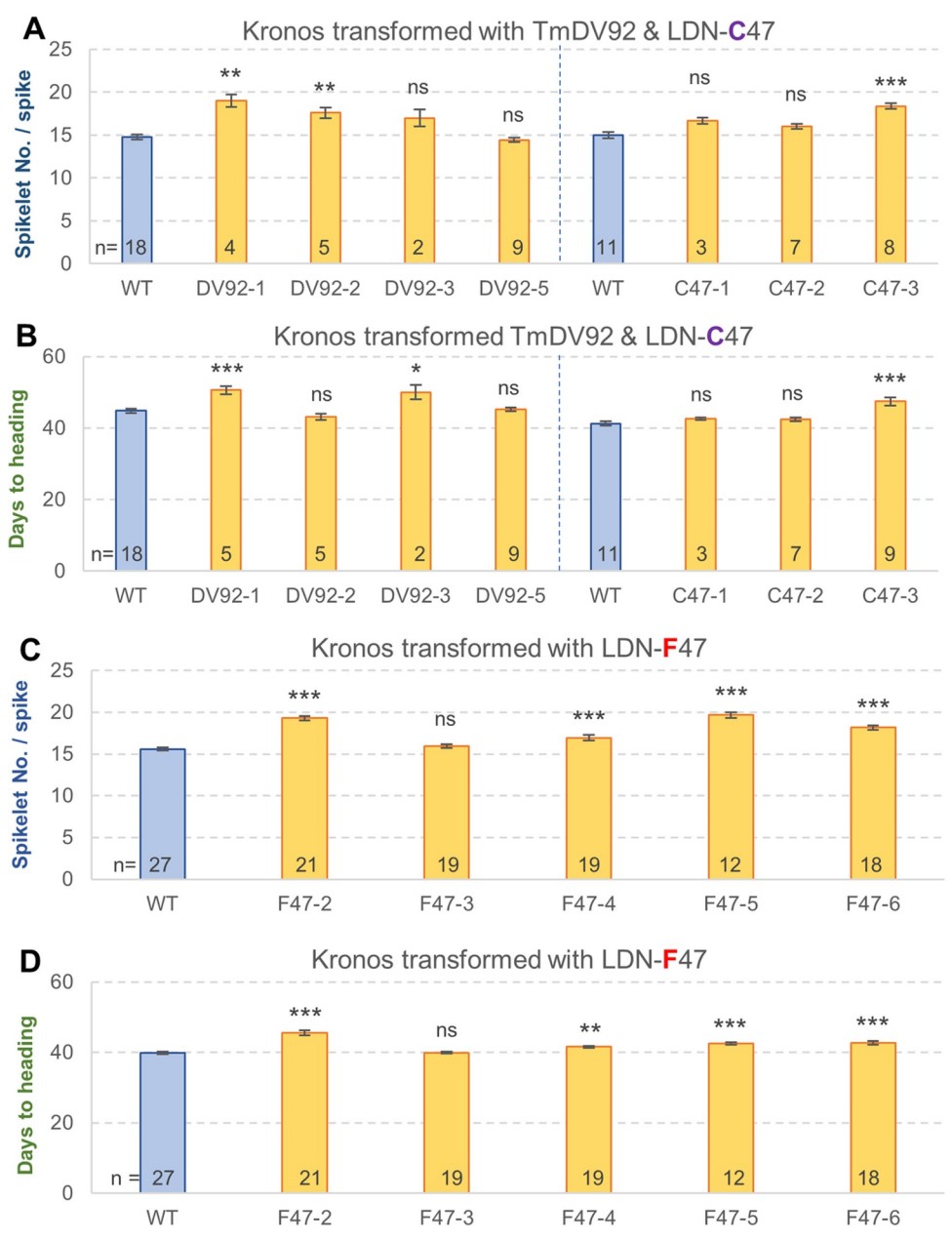

**Fig 2. Effect of transgenic *WAPO-A1* alleles on spikelet number per spike and heading date.** (A-B) Greenhouse experiment comparing four independent TmDV92 transgenic lines and three independent lines carrying the LDN-C47 allele with their respective non-transgenic sister lines (pooled from different events). (C-D) Separate greenhouse experiment comparing five independent transgenic lines carrying the LDN-F47 allele with the pool of non-transgenic sister lines. (A and C) Spikelet number per spike (only the spike from the main tiller was measured in each plant). (B and D) Heading date. Bars are averages and error bars are s.e.m. Numbers inside bars indicate number of plants and asterisks indicate *P* values (Dunnett test) relative to the pool of non-transgenic sister lines carrying the same construct. ns = not significant, * = $P < 0.05$, ** = $P < 0.01$ and *** = $P < 0.001$. Data used for this figure and the corresponding statistical analyses are available in Data B in S1 Data.

located at basal positions of the spike (Fig 4A and 4B). This abnormality was observed in transgenic plants of events TmDV92-1 (in 4 out of 6 $T_1$), LDN-C47-3 (in 3 out of 9 $T_1$), LDN-F47-2 (in 8 out of 21 $T_1$) and LDN-F47-6 (in 3 out of 18 $T_1$) (Figs 4A, 4B and S2). Transgenic line

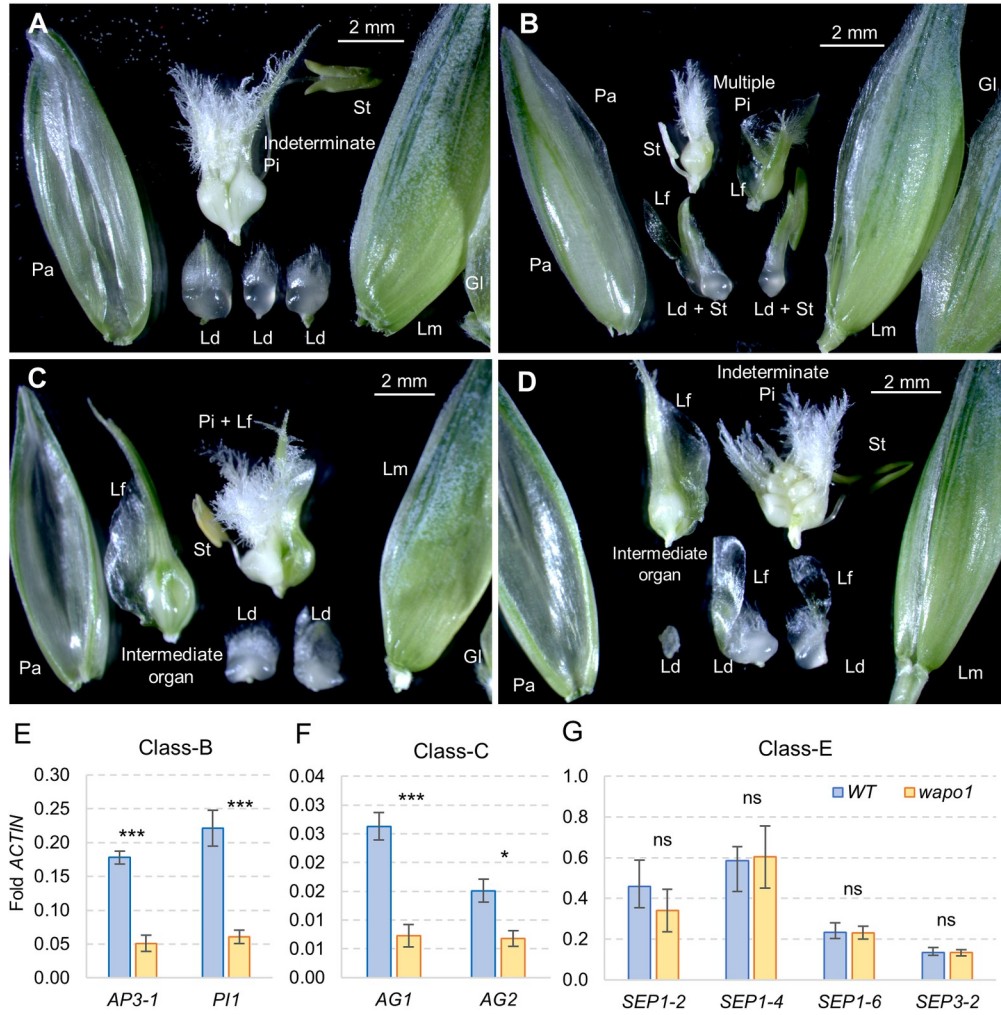

**Fig 3. Floral abnormalities and expression profiles of Kronos *wapo1* loss-of-function CRISPR mutant (*wapo-A1 wapo-B1*).** (**A**) Mutant $T_1$-101, floret one from basal spikelet showing three lodicules, one stamen and two fused pistils. (**B**) Mutant $T_1$-101, $2^{nd}$ floret from $3^{rd}$ most distal spikelet. Lodicules fused with stamens and leafy tissue. Indeterminate pistils fused with leafy tissue. (**C**) Mutant $T_1$-13, $2^{nd}$ floret of basal spikelet. One lodicule fused with leafy tissue, and an intermediate organ fused with leafy tissue, and pistil fused with leafy tissue. (**D**) Mutant $T_1$-13, $3^{rd}$ floret of basal spikelet. Lodicules with elongated leafy tissue, intermediate organ and indeterminate pistils. Lm = lemma, Pa = palea, Ld = lodicule, St = stamen, Pi = pistil, Lf = fused leafy tissue, + = fused organs. Frequency of the different floral abnormalities is available in Data C in S1 Data. A wild-type Kronos floret is shown in Fig 4 as reference. (**E-G**) Expression analysis of class-A, -B, -C and -E MADS-box genes in Kronos and *wapo1* mutants (MADS-box gene nomenclature is based on [20]). Primers for the qRT-PCRs are listed in S1 Table. Bars represent averages of four replicates and error bars are s.e.m. Each replicate is a pool of 12 developing spikes at the stamen primordia stage (~W4.0 in Waddington scale). ns = not significant, * = $P < 0.05$, ** = $P < 0.01$, and *** = $P < 0.001$. Data and statistical analyses used in E-G are available in Data D in S1 Data.

LDN-F47-2 had the most extreme phenotype showing multiple naked pistils, which were larger at the basal spikelets and transitioned to bracts in the more distal spikelets (S2A and S2B Fig).

A more frequent spike abnormality was the presence of smaller, densely packed spikelets in the distal region of the spike, ending in a small terminal spikelet. For simplicity, this spike abnormality is referred to hereafter as small terminal spikelet (Fig 4A red arrows and 4C). For event LDN-F47-3, which showed no significant differences in SNS, none of the $T_1$ plants had

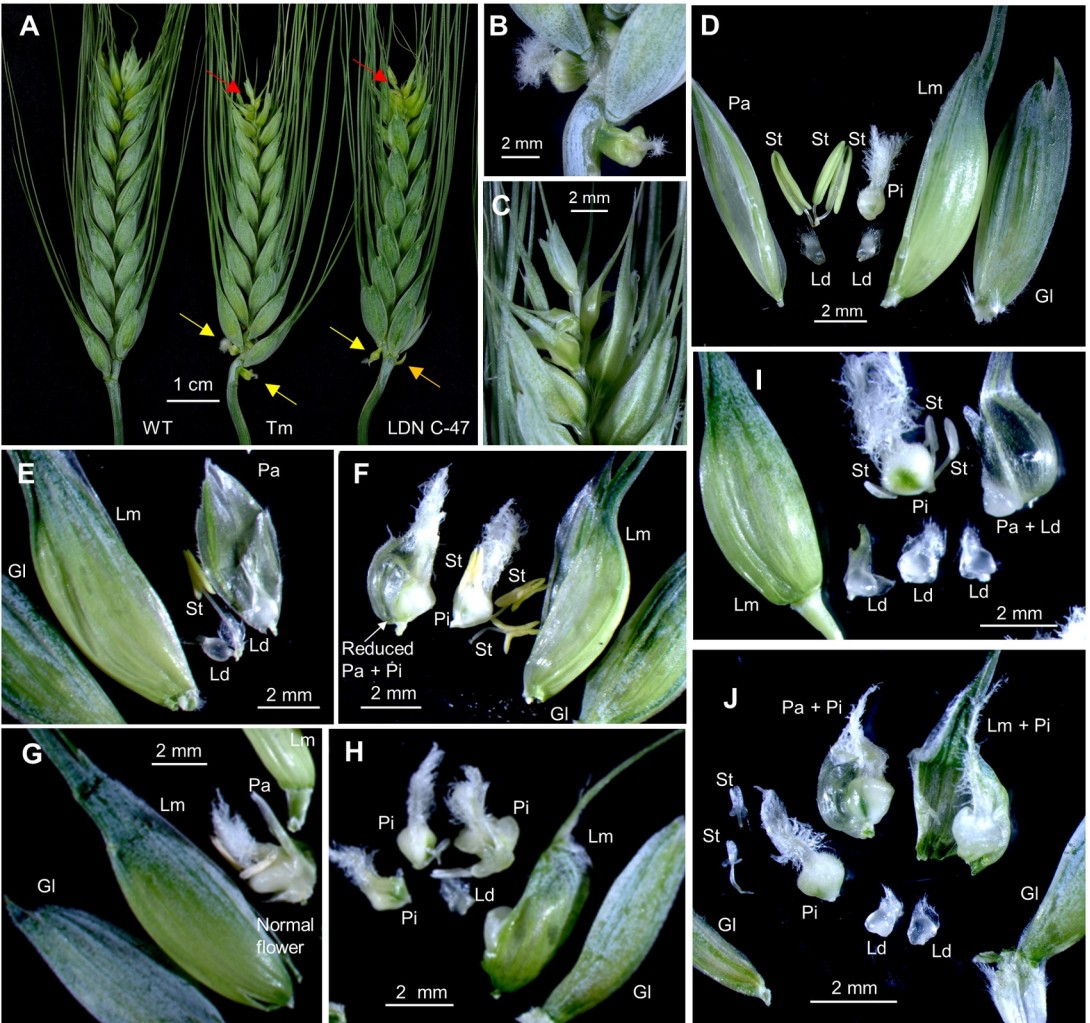

**Fig 4. Floral abnormalities in Kronos plants transformed with genomic regions of *WAPO-A1*.** (**A**) Spikes of non-transgenic control (WT) and transgenic lines TmDV92-1 and LDN-C47-3. (**B**) Detail of the naked pistils below the basal spikelets in TmDV92-1 (yellow arrows in A). (**C**) Detail of the abnormal distal end of the spike of TmDV92-1 (red arrows in A). (**D**) Normal floret from Kronos WT. (**E** and **F**) LDN-C47-3. (**E**) First floret in the 5th spikelet from the base: reduced palea, one stamen and no pistil. (**F**) Second floret in 11th spikelet from the base: no lodicules and extra pistils fused with small palea. (**G** and **H**) TmDV92-1. (**G**) First floret from basal spikelet: small palea and normal flower. (**H**) 17th spikelet from the base: one lodicule, two stamens and indeterminate pistils. (**I** and **J**) LDN-F47-2. (**I**) Third floret from basal spikelet: four lodicules, one fused to palea. (**J**) Spikelet 19th: two small stamens and three pistils (one fused to the palea and one to the lemma. Lm = lemma, Pa = palea, Ld = lodicule, St = stamen, Pi = pistil.

small terminal spikelets, whereas this phenotype was observed in 91.6% of the $T_1$ plants of LDN-F47-5, the event with the largest increase in SNS (26%). The other three LDN-F47 events, showed significant but smaller increases in SNS (9 to 24%) than the WT sister lines, and had small terminal spikelets in approximately half of the $T_1$ lines (47 to 50%).

Dissection of spikelets at different positions of the spike revealed increasing floral abnormalities towards the apical region. As reference, a normal floret from wild type Kronos is presented in Fig 4D. Examples of floral abnormalities are presented in Fig 4E and 4F (LDN-C47-3), Fig 4G and 4H (TmDV92-1), Fig 4I and 4J and S2 Fig (LDN-F47-2). The abnormal floral structures included (i) small paleae, (ii) variable numbers of lodicules, stamens and pistils per floret, and (iii) frequent fusions among these three organs and with lemmas and paleae

(Fig 4E–4J). Transgenic line LDN-F47-2, which had the most extreme phenotype for naked pistils (S2A and S2B Fig), also showed multiple floral defects. Spikelets near the terminal spikelet had higher numbers of pistils and reduced numbers of other organs (S2C Fig), whereas basal spikelets showed larger glumes, lemmas and stamens, particularly in the distal florets (S2D–S2G Fig).

For all three transgenic lines, the progeny of plants carrying the transgenes showed significantly higher ($P < 0.001$) transcript levels at W3.5 than their sister lines without the transgene (all plants have the endogenous H1 *WAPO-A1* allele) (Fig 5A–5C). In all three experiments, the progeny of plants from events with strong spike phenotypes (naked pistils and small terminal spikelet) showed higher *WAPO-A1* transcript levels (Fig 5A–5C), and stronger reductions in number of florets per spikelet (Fig 5D–5F) and average spikelet fertility (Fig 5G–5I) than the progeny of the events with weaker spike phenotypes and intermediate *WAPO-A1* expression levels. The progeny of F47-2-1 segregated for weak and strong spike phenotypes and the

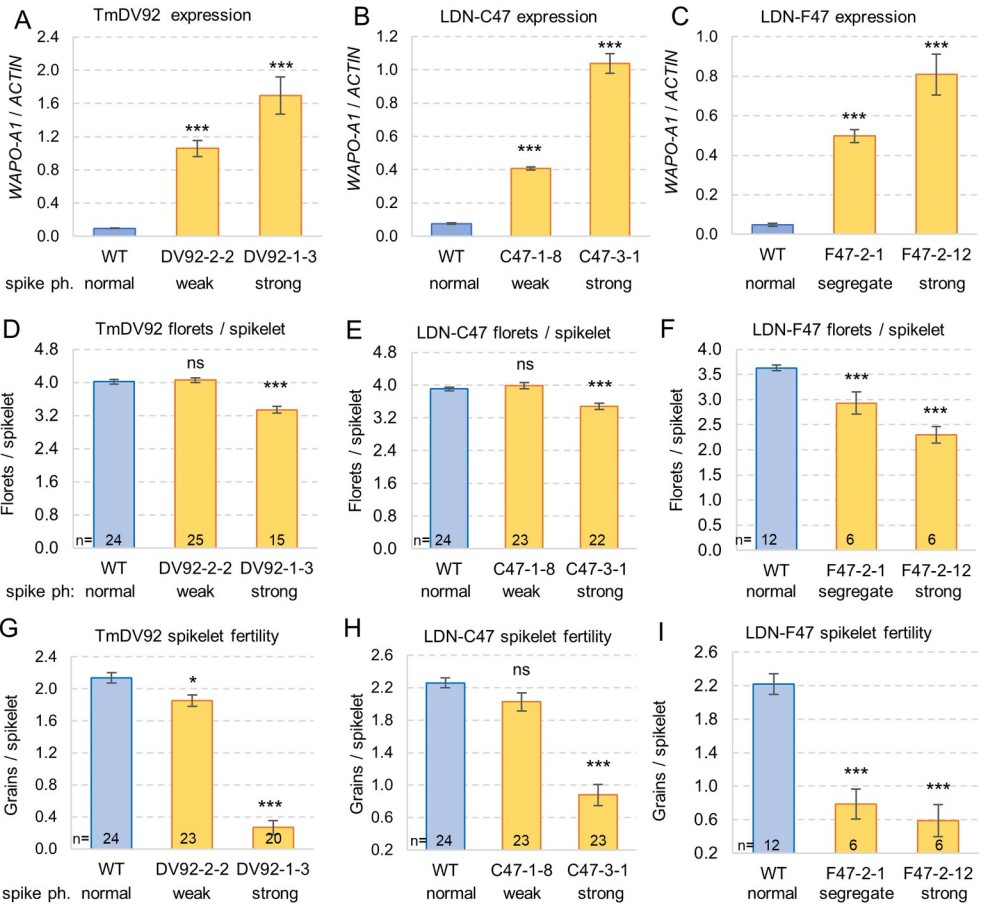

**Fig 5. *WAPO-A1* transcript levels and fertility in Kronos transgenic plants.** (**A–C**) qRT-PCR transcript levels calculated with the ΔCt method using *ACTIN* as an endogenous control. Bars are averages of four replications and each replication corresponds to a pool of 6–9 developing spikes at the floret primordia stage (W3.5). Spike ph. = spike phenotype, weak = differences in SNS, strong = naked pistils and small terminal spikelet, segregate = segregation for strong and weak spike phenotypes (F47-2-1). (**D–F**) Average floret number per spikelet (**G—I**) Spikelet fertility calculated as GNS / SNS. Only one spike per plant was measured. Kronos plants transformed with (**A, D** and **G**) TmDV92 (from *T. monococcum*). (**B, E** and **H**) LDN-C47, and (**C, F** and **I**) LDN-F47. The first number in a transgenic plant designation corresponds to the transformation event. ns = not significant, * = $P < 0.05$, *** = $P< 0.001$, indicate Dunnett test *P* value relative to the WT sister line. Bars represent averages and error bars are s.e.m. Numbers at the base of the bars indicate plants analyzed. Data and statistical analyses are available in Data E in S1 Data.

number of florets per spikelet and spikelet fertility was more affected than in the progenies of the other two constructs with intermediate levels of *WAPO-A1* expression. In all three constructs, the reductions in spikelet fertility were stronger than the reductions in the number of florets per spikelet, suggesting that both reduced floret fertility (due to floral defects) and reduced number of florets per spikelet (due to smaller distal spikelets) contributed to the drastic reductions in spikelet fertility and GNS in the transgenic plants with high *WAPO-A1* expression.

## *WAPO1* is expressed in inflorescence, spikelet and floral meristems

To better characterize the expression profile of *WAPO1*, we investigated its spatial and temporal distribution during spike and spikelet development in Kronos (Fig 6) and diploid *T. monococcum* (S3 Fig) by *in situ* hybridization. In both species, weak expression of *WAPO1* was detected in the inflorescence meristem (IM) at the double ridge stage (W2.5, Figs 6A and S3A) and in both the IM and spikelet meristems (SM) at the subsequent W3.0 stage (Figs 6B and S3B). At W3.25 -W3.5, a strong expression of *WAPO1* was detected in the incipient floret meristems (Figs 6C and S3C). In summary, *WAPO1* expression in the inflorescence and spikelet meristems correlates well with its effects on the timing of terminal spikelet induction, whereas its expression in the floret meristem correlates with its role in floral development and the flower defects observed in *WAPO1* mutants and transgenic plants.

## Natural variation in *WAPO-A1* is associated with changes in SNS

In addition to the validation of *WAPO-A1* as the causal gene for the 7AL SNS QTL, we investigated the effect of the different *WAPO-A1* alleles on SNS and GNS. In our previous study, we

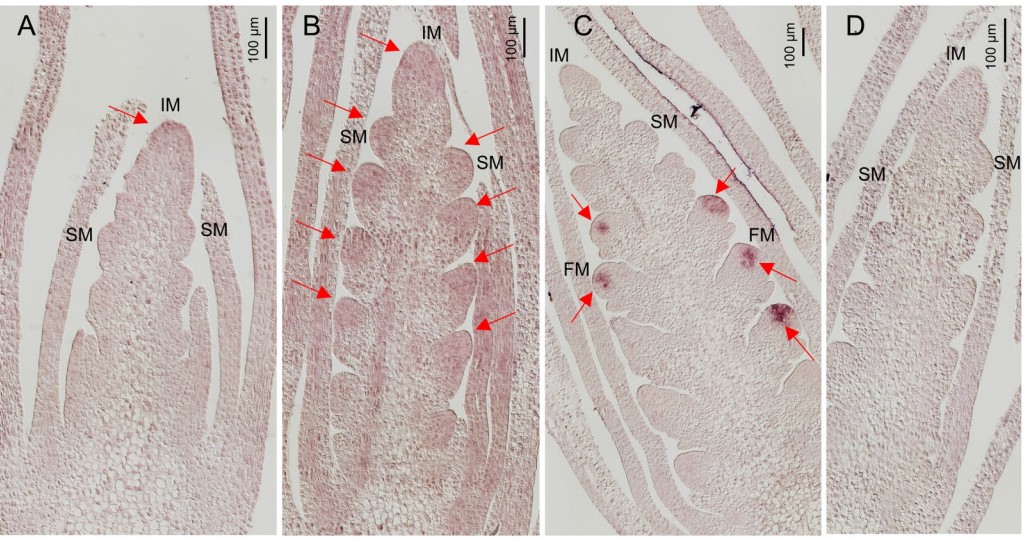

*WAPO1* antisense    *WAPO1* sense

**Fig 6.** ***In situ* hybridization analysis of *WAPO-A1* in sections of developing spikes of tetraploid wheat Kronos.** (**A**-**C**) *WAPO1* antisense probe. (**D**) *WAPO1* sense probe used as a negative control in a developing spike at W2.5 with undifferentiated spikelet meristems (SM). (**A**) Developing spike at the double ridge stage (W2.5). (**B**) Early spike development with undifferentiated spikelet meristems (W3.0). Weak *WAPO1* expression was detected in the inflorescence meristem and spikelet meristem. (**C**) Spikes at W3.25-W3.5 showing strong expression of *WAPO1* in the incipient floret meristems of the more developed spikelets closer to the center of the spike. W values indicate developmental stages based on the Waddington scale. IM = inflorescence meristem, SM = spikelet meristem, and FM = floret meristem. Red arrows indicate *WAPO-A1* hybridization signals.

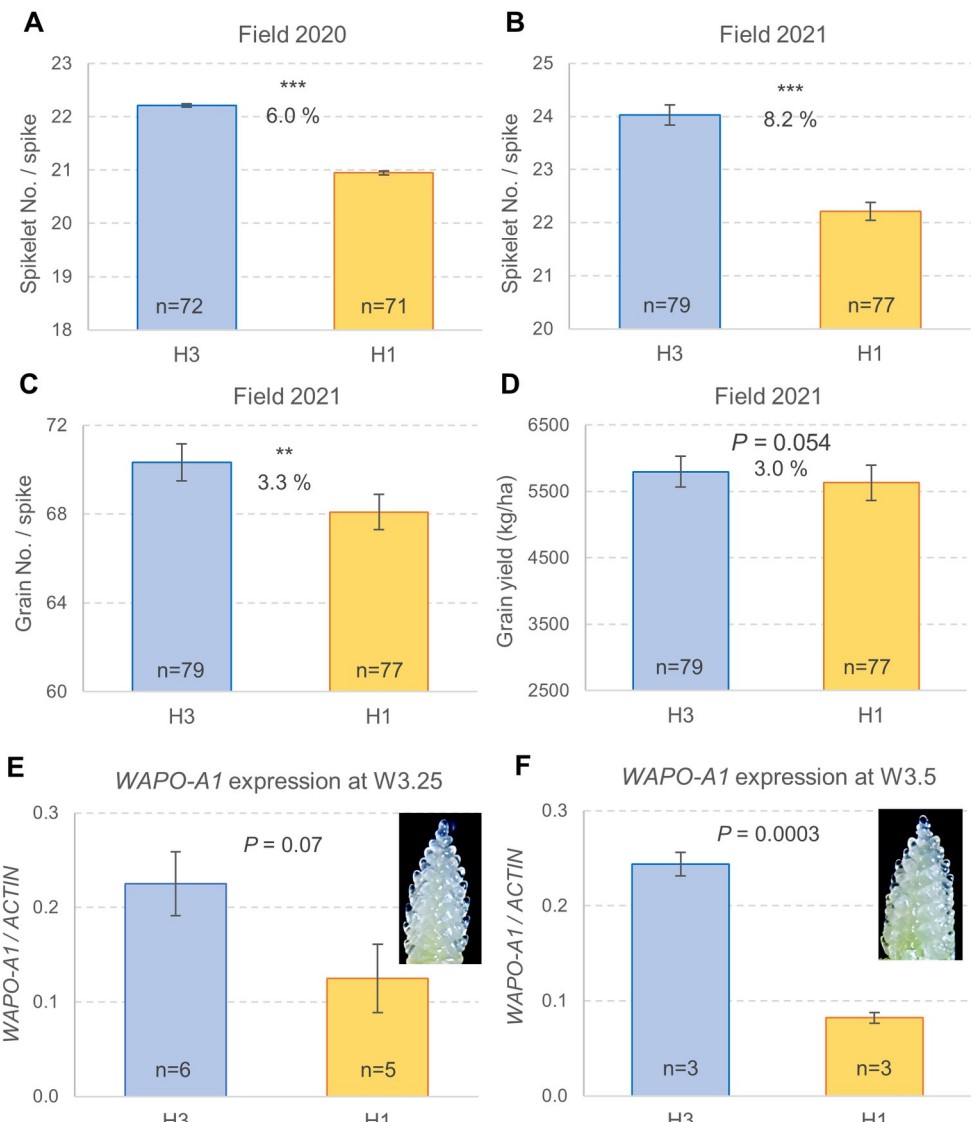

**Fig 7. Effect of *WAPO-A1* haplotypes H1 and H3 on SNS and *WAPO-A1* transcript levels.** (**A**) 2020 RCBD-split plot field experiment. (**B-D**) 2021 RCBD split-plot field experiment. (**B**) Spikelet number per spike. (**C**) Grain number per spike. (**D**) Grain yield (kg/ha) (**E-F**) Independent qRT-PCR experiments comparing H1 and H3 transcript levels of *WAPO-A1* relative to *ACTIN* using the $2^{\Delta Ct}$ method. (**E-F**) Pools of 3–8 developing spikes from the main tiller. (**E**) Collected when lemma primordia were present (W3.25). (**F**) Collected when floret primordia were present (W3.5). Bars represent averages and error bars are s.e.m. Numbers inside the bars indicate the number of replications. ns = not significant, * = $P < 0.05$, ** = $P < 0.01$ and *** = $P < 0.001$. Data used for this figure are available in Data G in S1 Data, and the statistical analyses in S3 Table.

established that the *WAPO-A1* allele in the H2 haplotype was associated with higher SNS than the H1 and H3 haplotypes [10], but we did not determine the relative effects of the H1 and H3 haplotypes on SNS. To address this question, we performed two field experiments in 2020 (Fig 7A and panel A in S3 Table) and 2021 (Fig 7B–7D and panels B-G in S3 Table), both organized in an RCBD-split plot design. Both experiments had 10 replications, each including eight $F_{4:6}$ heterozygous inbred families (HIF) as main plots and homozygous H1 and H3 sister lines as subplots. Average SNS for plants homozygous for the H3 haplotype was 6% higher than for those homozygous for the H1 haplotype in the 2020 experiment (1.3 spikelets) and 8.2% higher

in the 2021 experiment (1.8 spikelets). The overall differences in SNS were highly significant both years ($P < 0.0001$, Fig 7A and 7B, panels A and B in S3 Table). Within individual families, differences in SNS between H1 and H3 were significant for six out of the 8 families in 2020 and for all families in 2021 (panels A and B S3 Table).

In the 2021 experiment, we had sufficient grains to use small plots (1.86 m$^2$) as experimental units and to provide a preliminary estimate of grain yield. In this experiment, the H3 haplotype showed a 3.3% increase in GNS ($P = 0.0085$, Fig 7C and panel C in S3 Table). The increase in GNS was smaller than the increase in SNS (8.2%) because it was partially offset by a 4.6% decrease in fertility (grains per spikelet) in H3 relative to H1 ($P < 0.001$, panel D in S3 Table and Data F in S1 Data). The H3 haplotype was also associated with a 1.6% decrease in kernel weight ($P = 0.0281$, panel E in S3Table) relative to H1. The balance of these opposite effects was a 3.0% increase in grain yield, but the difference was marginally not significant ($P = 0.0545$, Fig 7D and panel F in S3 Table). In this experiment, we did not detect significant differences in heading time ($P = 0.5987$, panel G in S3 Table).

To test if the differences in SNS between H1 and H3 haplotypes were associated with differences in the expression levels of *WAPO-A1*, we performed three independent qRT-PCR experiments comparing homozygous H1 and H3 sister lines derived from $F_{4:6}$ HIF #120. Developing spikes from the main tiller were collected at the lemma primordia stage (W3.25) for the first two experiments and at the floret primordia stage (W3.5) for the third experiment (Fig 7E and 7F). The first two experiments at W3.25, which were analyzed together using experiment as block, showed 1.8-fold higher transcript levels in H3 than in H1, but the difference was marginally not significant ($P = 0.07$, Fig 7E and Data G in S1 Data). In the third experiment at W3.5, transcript levels in H3 were 3.0-fold higher than in H1 and the differences were highly significant ($P = 0.0003$, Fig 7F and Data G in S1 Data).

In the same field used for comparing the H1 and H3 haplotypes in 2021, we also compared the relative effects of the H2 and H1 haplotypes in near isogenic lines (NILs) of tetraploid wheat Kronos and hexaploid wheat GID4314513 (a high-biomass line from CIMMYT). In the Kronos $BC_3F_3$ NILs, the H2-NILs showed 17.7% more spikelets per spike ($P < 0.0001$, Fig 8B) and 6.7% more grains per spike ($P = 0.025$, Fig 8C) than the H1-NILs (panels A-B in S4 Table). The lower percent increase in GNS than in SNS can be partially explained by a 9.2% reduction in spikelet fertility (GNS/SNS) in the H2-NILs ($P < 0.0001$, Fig 8C and panel C in S4 Table). Grain weight per spike (grain number x grain weight) was 7.1% higher in the H2-NILs than in the H1-NILs, but the differences were not significant ($P = 0.11$, Fig 8D and panel D in S4 Table). The H2-NILs were also associated with a non-significant increase in grain weight (0.4%) and a non-significant delay in heading time (0.7% panels E-F in S4 Table). Total grain yield was not estimated in this experiment because experimental units were single rows.

In the hexaploid $BC_5F_3$ NILs, tested in small plots, we detected a 6.1% increase in SNS in the H2-NILs relative to the H1-NILs ($P = 0.0092$, Fig 8E and panel G in S4 Table) and an 8.4% increase in grain yield that was marginally significant ($P = 0.0497$, Fig 8F and panel H in S4 Table). The H2 haplotype was also associated with non-significant differences relative to H1 for GNS (+2.7%), fertility (-3.3%), kernel weight (+1.8%) and grain weight per spike (+4.4%, panels I-L in S4 Table).

## Discussion

In our previous study, we identified *WAPO-A1* as the leading candidate gene for the 7AL SNS QTL based on two high resolution genetic maps [10]. In this study, we demonstrate that *WAPO1* is both necessary and sufficient to increase SNS in wheat and, therefore, that *WAPO-A1* is the causal gene underpinning the 7AL QTL for SNS.

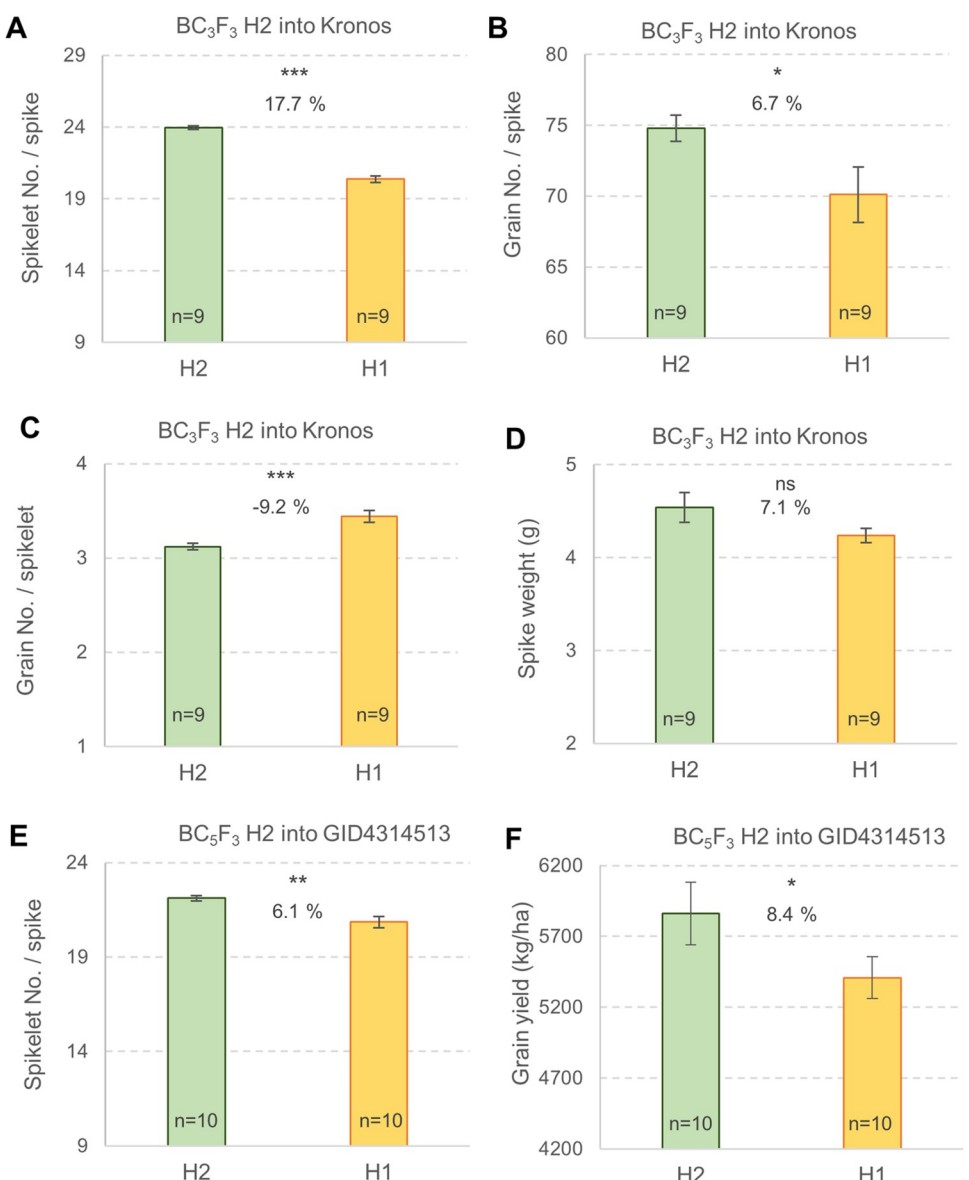

**Fig 8. Effect of *WAPO-A1* haplotype H2 introgressed into tetraploid Kronos and the high-biomass hexaploid line GID4314513 (both H1).** Experiments performed in the field in 2021. (**A-D**) Homozygous BC₃F₃ sister lines (CRD, n = 9, 1-m rows, 10 spikes measured per row). (**A**) Spikelet number per spike. (**B**) Grain number per spike. (**C**) Grain number per spikelet (spikelet fertility). (**D**) Grain weight per spike. (**E-F**) Homozygous BC₅F₃ sister lines (RCBD, n = 10, 1.86 m² plots, 4 spikes measured per plot). Error bars are s.e.m. ns = not significant, * = P < 0.05, ** = P < 0.01, *** = P < 0.001. Data used to generate this figure are available in Data H in S1 Data and the statistical analyses in S4 Table.

## Potential mechanisms involved in *WAPO1* effect on SNS

Differences in *WAPO-A1* transcript levels in different genotypes correlate well with differences in SNS. The reduced expression of the *WAPO-A1* allele (115-bp promoter deletion) in H1 relative to the H2 [10] and H3 haplotypes (Fig 7) is associated with lower SNS in H1 relative to the other two haplotypes. Similarly, the higher transcript levels of *WAPO-B1* relative to *WAPO-A1* during spike development (S1 Fig) is associated with stronger effects of the *wapo-B1* mutation on SNS relative to the *wapo-A1* mutation (Fig 1D). These results, together with the partial

codominant effect of *WAPO1* on SNS (Fig 1E) and the increased SNS in transgenic plants with additional genomic copies of *WAPO-A1* (Fig 2), support the hypothesis that higher *WAPO1* transcript levels can drive increases in SNS.

Increases in SNS are associated with a delay in the transition of the inflorescence meristem (IM) into a terminal spikelet, and our *in situ* hybridization results confirmed that *WAPO1* is weakly expressed in the IM at the double ridge stage in both diploid (S3A Fig) and tetraploid wheat (Fig 6A). During the early stages of spikelet development, we detected hybridization signals in the IM and the spikelet meristems (SM, Fig 6B). As spike development approached the formation of the terminal spikelet and the lateral spikelets started to differentiate, the *WAPO1* hybridization signal was no longer detected in the IM and was replaced by a strong signal in the early floret meristems (FM, Fig 6C).

The barley ortholog of *WAPO1*, *HvAPO1*, has been also detected in the IM and FM of the developing barley spikes by *in situ* hybridization [21]. Those experiments showed a strong *HvAPO1* hybridization signal in both the IM and FM at the time of spikelet differentiation (see Fig S10 in reference [21]). By contrast, at the time when *WAPO1* was highly expressed in the FM in the wheat developing spike (W3.25-W3.5), we failed to detect *WAPO1* in the IM (Figs 6C and S3C). It would be interesting to investigate if the stronger and extended expression of *HvAPO1* in the barley IM during floret development contributes to the different fate of the IM in barley (indeterminate inflorescence) compared to wheat (determinate inflorescence).

In rice, *APO1* is also expressed in the IM and in the primary and secondary branch meristems of the panicle, and *apo1* mutants have a reduced number of branches and spikelets [11]. In rice, it was shown that APO1 physically interacts with LFY (APO2), and that the two genes act cooperatively to control spikelet number per panicle [15]. Similar *in vitro* and *in vivo* interactions have been reported between the Arabidopsis homologs UFO and LFY [14,22]. Mutants for both genes show similar defects in inflorescence structure and morphology, further suggesting that UFO and LFY act cooperatively to promote floral-meristem identity [23]. Based on the rice and Arabidopsis results, we hypothesize that *LFY* may participate in the mechanism by which *WAPO1* regulates SNS in wheat. In Arabidopsis, *LFY* has been shown to act both as a transcriptional activator of the *SQUAMOSA* MADS-box gene *AP1* [24] and as a repressor of the *SVP* MADS-box genes *AGL24* and *SVP* [25]. Therefore, it would be interesting to investigate the roles of *WAPO1* and *LFY* in the regulation of the wheat *AP1* homologs (*VRN1*, *FUL2* and *FUL3*) and *SVP* homologs (*VRT2* and *SVP1*), which have been previously shown to be involved in the regulation of SNS in wheat [26,27].

## Floral and spike abnormalities in mutant and transgenic *WAPO-A1* plants

In addition to its role in the activation of *AP1*, LFY has been shown to bind to regulatory regions of *APETALA3* (*AP3*, a class-B floral gene) and *AGAMOUS* (*AG*, a class-C floral genes) in Arabidopsis [14,28–30]. The activation of *AP3* expression requires the activity of both *LFY* and *UFO*, which likely explains the reported downregulation of class-B genes in the *ufo* mutant in Arabidopsis [13,14] and class-C genes in the *apo1* mutant in rice [12]. In the developing spikes of the wheat *wapo1* null mutants, we also observed downregulation of class-B and -C but not class-E genes relative to the WT, suggesting a conserved molecular mechanism.

Class-B and -C MADS-box genes in Arabidopsis are associated with petal, stamen and pistil identity, which explains the stronger abnormalities detected in the *ufo* mutants in these three inner floral whorls. It has been shown that ectopic expression of both *PI* and *AP3* rescues the floral organ identity defects of the Arabidopsis *ufo* mutant [31]. Stronger floral defects in the inner whorls of the flowers than in glumes and lemmas have been observed before in the rice *apo1* mutant [12], and in this study in the wheat *wapo1* mutant. The related floral defects

observed in the *ufo*, *apo1* and *wapo1* mutants suggest functional conservation of these homologous genes across the monocot eudicot divide.

The strong expression of *WAPO1* in the floret meristem (Figs 6C and S3C), supports the critical role of this gene in wheat floral development. Transgenic wheat plants with additional *WAPO1* genomic copies shared some floral aberrations with the *wapo1* mutant, including differences in the number of lodicules, stamens and pistils and fusion among organs. In addition, mutants and transgenic plants with strong floral abnormalities showed reduced fertility (Fig 5G and 5I). These results suggest that decreases or increases in *WAPO1* expression can lead to abnormal development in the three inner whorls of the floret. Transgenic *WAPO-A1* plants showed additional phenotypes not observed in the *wapo1* mutant, including reduced and sometimes curved paleae (homologous to sepals), a compact spike tip with a small terminal spikelet, and a significant delay in heading time.

A less frequent but unusual phenotype in the transgenic *WAPO1* plants was the presence of naked pistils. A similar phenotype has been described in the Arabidopsis *ufo* mutants, where structures resembling normal pistils were formed at the end of primary and coflorescence shoots [13,32]. The naked pistils in the transgenic wheat plants appeared adjacent to and directly below normal spikelets, in a position similar to the paired spikelets observed in loss-of-function mutants for *ppd-D1* and *ft-B1* in hexaploid wheat [33]. However, the paired spikelets in the previous two mutants were more frequent in the central part of the spike, whereas in the *WAPO-A1* transgenic plants the naked pistils were more frequent in the basal region of the spike. Based on the terminal position of the naked pistils in Arabidopsis *ufo* mutants, we hypothesize that the naked pistils in the transgenic wheat plants may be located at the end of a short branch including one lateral spikelet. Branches in the lower nodes of the wheat spike have been reported in the *Branched head* mutants, which carry mutations in an *AP2/ERF* transcription factor [34,35].

## Effects of different *WAPO-A1* natural alleles and their potential applications in wheat breeding

Our previous study showed that the H2 haplotype was associated with larger increases in SNS than the H1 and H3 haplotypes in several segregating populations [10]. The favorable effect of H2 on SNS was observed in spring and winter wheats and in different wheat market classes, but these increases did not always result in higher grain yield. In genotypes or environments with insufficient resources to develop and/or fill the extra grains, increases in SNS were partially or completely offset by decreases in fertility and/or grain weight. However, when H2 was present in well adapted and productive genotypes, and the plants were grown in favorable environments, the increases in SNS were associated with significant increases in grain yield [10]. In this study, we also observed significant 8.4% increases in total grain yield when the H2 haplotype was introgressed into a highly productive hexaploid line from the high-biomass program from CIMMYT and the plants were grown under favorable irrigation and fertilization conditions. This is a promising preliminary result, but needs to be further validated with larger yield trials in different genetic backgrounds.

The positive effects of the H2 haplotype on SNS in the less productive tetraploid wheat Kronos was partially offset by a reduction in spike fertility (-9.2%), which resulted in a smaller increase in GNS (6.7%) than in SNS (17.7%). In this experiment we did not observe a negative effect of the H2 haplotype in grain weight. The balance of these different effects was a positive 7.1% increase in grain yield per spike, but this difference was not significant ($P = 0.1078$). Based on these results, we hypothesize that when the plant has more available spikelets, it adjusts the number of grains or their filling to the available resources, depending on the timing during development when the plant faces the limited resources.

An increase in the frequency of the H2 haplotype was previously described in common wheat, from less than 50% in old landraces to more than 80% in modern wheat varieties [10]. This suggests that selection for high SNS, GNS, or grain yield may have accelerated the increases in H2 frequencies. By contrast, in tetraploid wheat the H1 haplotype replaced the H3 haplotype and became almost fixed in modern durum varieties (99%) [10]. This increase in H1 frequency cannot be explained by selection for higher SNS or GNS, because our field experiment showed that H1 has significantly lower SNS and GNS than H3 (Fig 7). Since durum wheat has much larger grains than common wheat, we speculate that selection for larger grains may have resulted in an indirect selection for reduced grain number, favoring the H1 allele. This indirect selection was likely driven by the negative correlation between grain number and weight, a trade off that is magnified in source-limited varieties and/or suboptimal environments.

A similar result has been recently reported for the *FT-A2* gene, where the derived A10 allele associated with increases in SNS was found at high frequency in common wheat (~60%) but was almost absent in modern durum wheat commercial varieties (0.7%) [36]. These results suggest that other genes involved in the regulation of spike development may have differential frequencies between common and durum wheat as a result of this putative selection for large grains.

However, it is also possible that the derived *WAPO-A1* H2 and *FT-A2* A10 alleles were never properly tested in modern durum wheat varieties. The *WAPO-A1* H2 haplotype was found in only one accession from Syria in a survey of 364 durum wheat varieties [10]. Similarly, the *FT-A2* A10 allele was found only in three accessions from Oman and Turkey in a survey of 417 durum wheat varieties [36]. In this study, we show that the transfer of the *WAPO-A1* H2 haplotype from common wheat into the tetraploid wheat Kronos was associated with significant increases in SNS and GNS. A similar favorable effect of H2 relative to H1 on SNS was previously reported in a segregating population generated from a cross between cultivated emmer and durum wheat in field trials in Fargo (ND) [10]. These are promising results, but they still require validation in different durum varieties and larger yield trials. To facilitate the introgression of the H2 haplotype in durum wheat, we deposited the Kronos NIL with the H2 introgression in the National Small Grains Collection (PI 698810).

In summary, the validation of *WAPO-A1* as the causal gene of the 7AL SNS QTL and of the positive effects of the H2 haplotype on spikelet and grain number per spike, provides wheat breeders a new tool to improve sink traits with limited effects on heading time. We hypothesize that the introgression of the H2 haplotype into varieties that are not limited in source traits (e.g. varieties with high biomass) can result in increases in total grain yield in favorable environments.

## Materials and methods

### Ethyl methanesulfonate (EMS) induced mutations in *WAPO-A1*

We screened a database of sequenced EMS induced mutations in the tetraploid wheat variety Kronos [16] using the sequence of *WAPO-A1* (*TraesCS7A02G481600*). Kronos has the *WAPO-A1* H1 haplotype, which carries a 115-bp deletion in the promoter and a change from aspartic acid to asparagine at position 384 (D384N). For the A-genome homeolog (*WAPO-A1*), we identified line K4222 carrying a mutation that results in the replacement of a tryptophan by a premature stop codon at position 216 of the protein (W216*). For the B-genome homeolog (*WAPO-B1*), we identified five mutations that resulted in amino acid changes but none generated premature stop codons or altered splicing sites; hence, our first studies of the EMS mutants only included K4222. We crossed the K4222 mutant to the non-mutagenized Kronos, self-pollinated the $F_1$, and selected sister lines homozygous for the

mutant (*wapo-A1*) and the WT alleles (*Wapo-A1*) from the segregating $F_2$ plants. The derived $F_3$ plants were evaluated in the greenhouse and the $F_4$ plants in the field. To further reduce background mutations, we generated $BC_1F_2$ homozygous sister lines by backcrossing the $F_1$ with Kronos, self-pollinating the $BC_1$, and selecting homozygous plants in the next generation. The derived $BC_1F_3$ grains were planted in a field experiment.

For the greenhouse experiment, we used two $F_3$ plants per pot (3.8 L) and measured average SNS per plant. In the first field experiment, we planted homozygous mutant and WT $F_4$ sister lines in a complete randomized design (CRD) with 10 replicates per genotype. Each replicate included a row of 2–5 plants spaced 0.3 meters apart. Average SNS per row was calculated by measuring SNS from 3 spikes per plant. In the second field experiment, we planted $BC_1F_3$ plants 0.3 m apart segregating for the *WAPO-A1* alleles. Genotyping of these plants revealed 11 homozygous *wapo-A1* and 8 homozygous WT within the same $BC_1$ family #54. For each plant, we determined the average SNS from four spikes. Both field experiments were conducted at the University of California Experimental Field Station in Davis, referred to hereafter as UC Davis, during the months of October 2019 and June 2020.

## Generation of *wapo-A1 wapo-B1* double mutants using CRISPR-Cas9

To characterize better the function of *WAPO1*, we edited both homeologs of the tetraploid variety Kronos using CRISPR-Cas9 [37]. We designed one guide RNA between positions 494 and 512 from the starting ATG of the coding region (S1 Table) to induce double-strand breaks in the first exon of both *WAPO-A1* and *WAPO-B1*. This guide RNA was then cloned into a vector which included the Cas9 gene and a *GRF4-GIF1* chimera that increases wheat regeneration efficiency for *Agrobacterium*-mediated transformation [38]. Three independent $T_0$ transgenic Kronos plants were obtained from the UCD Plant Transformation facility and were screened for mutations by next generation sequencing (NGS) and restriction enzyme digestion. For the NGS screen, we used primers g641-NGS-F and R that amplify both genomes (S1 Table) and analyzed the data using CRISgo (https://github.com/pinbo/CRISgo) as published before [39]. For the restriction enzyme screen, we used A-genome specific primer pair CAPS-*WAPO-A1*-F and R1 and B-genome specific primer pair CAPS-*WAPO-B1*-F and R2 (S1 Table) followed by digestion with restriction enzyme *Xcm*I. The same primers and restriction enzyme digestion were used as Cleavage Amplified Polymorphism Sequence (CAPS) markers for subsequent generations.

## Generation of *WAPO1* transgenic lines

To validate the role of *WAPO-A1* in the regulation of SNS, we generated transgenic plants expressing the *WAPO-A1* gene driven by its native promoter. The genomic region of *WAPO-A1*, which included 4.8 kb upstream of the start codon, the complete coding region with its intron, and 1.5 kb downstream of the stop codon, was cloned into the hygromycin-resistance binary vector pLC41. Three different *WAPO-A1* alleles, each encoding a different WAPO-A1 protein (Table 1), were cloned and transformed into Kronos. The first genomic region of *WAPO-A^m1* was obtained from the BAC library of diploid *T. monococcum* ssp. *monococcum* accession DV92, henceforth TmDV92 (genome A^m) [18]. The WAPO-A^m1 protein encoded by the TmDV92 allele differs from WAPO-A1 in polyploid wheat by three amino acids, but is otherwise similar to the H3 haplotype (Table 1). The second *WAPO-A1* genomic region was cloned from the BAC library of *T. turgidum* ssp. *durum* variety Langdon [19]. This construct, designated hereafter as LDN-C47, encodes the *Wapo-A1d* allele (H3 haplotype), which has no deletion in the promoter region and has the ancestral amino acids C47 and D384 (Table 1)

The third construct, named LDN-F47, was created by site-directed mutagenesis of construct LDN-C47 using primers described in S1 Table. The change from the ancestral cysteine (C47) into a phenylalanine (F47), resulted in a protein identical to the one encoded by the *Wapo-A1b* allele in the H2 haplotype (Table 1). However, the LDN-F47 clone differs from the natural H2 genomic region by three SNPs in the promoter region and two in the first intron [10]. Table 1 presents a summary of the different *WAPO-A1* constructs and their comparison with the endogenous *Wapo-A1a* allele (H1 haplotype) present in the transformed variety Kronos.

Constructs were transformed into Kronos using *Agrobacterium*-mediated transformation (EHA105) at the UC Davis Plant Transformation Facility as described before [38]. Primers listed in S1 Table were used to genotype the transgenic plants and confirm the presence of the transgene. Transgenic $T_0$ plants were advanced to $T_1$ and characterized in the greenhouse for heading date, SNS, and spike and floral morphology.

## Populations used to compare the effects of *WAPO-A1* haplotypes in the field

To compare the effect of the H1 and H3 haplotypes on SNS, we developed a population segregating for the H1 haplotype from Kronos and the H3 haplotype from the durum wheat variety Rusty, which carried the same *Wapo-A1d* allele as Langdon [10] (Table 1). From the Kronos x Rusty cross [40], 75 $F_2$ plants were advanced to $F_4$ by single seed descent (SDS) and genotyped for the *WAPO-A1* haplotype using a molecular marker previously developed for the *WAPO-A1* promoter deletion [10]. We selected eight heterozygous $F_4$ plants (IDs = 12, 19, 51, 55, 69, 113, 120, 128) to generate eight Heterozygous Inbred Families (HIFs). For each $F_{4:5}$ HIF, we selected two homozygous sister lines ($F_{4:6}$)–one fixed for H1 and the other fixed for H3.

The $F_{4:6}$ grains were used for two field trials conducted at UC Davis experimental field station that were planted in October and harvested in June of the next year. The experiments are referred to using their harvest years (2020 and 2021). Both field experiments were organized in a split plot randomized complete block design (RCBD) with 10 replications, using the eight HIFs as main plots, and sister lines fixed for H1 and H3 haplotypes as subplots. For the 2020 experiment, we used single rows as experimental units and measured 15 spikes per row. For the 2021 experiment, we used small plots (3 rows = 1.86 $m^2$) as experimental units and measured 12 spikes per plot for SNS determinations. In 2021, we also measured heading date as the time from the first rain after sowing to the time when half of the plants in the plot have headed, grain number per spike (each replication was the average of 3 spikes), and total grain yield in kg/ha (small plots were harvested with a Wintersteiger Combine).

To compare the effect of H2 relative to H1, we developed NILs segregating for these haplotypes in both tetraploid and hexaploid wheat. The tetraploid NILs were developed by introgressing the *WAPO-A1* H2 haplotype from UC Davis common wheat breeding line UC1110 (also referred to as CAP1) into tetraploid wheat variety Kronos using marker assisted backcrossing. After three backcrosses with Kronos, we self-pollinated the $BC_3$ plants, and selected $BC_3F_2$ plants homozygous for the H1 and H2 haplotypes. The derived $BC_3F_3$ grains were sown at UC Davis in November 2020 in a completely randomized design (CRD) with nine replications, using 1-m rows as experimental units (10 spikes were measured per row and averaged). The hexaploid NILs were developed by backcrossing the H2 haplotype five times into the hexaploid line GID4314513 (H1) from the CIMMYT high-biomass program. We selected a high-biomass line as recurrent parent to increase the probability that increases in SNS and GNS would be translated into increases in total grain yield. We developed $BC_5F_3$ homozygous NILs

using GID4314513 as recurrent parent, and compared them in the field at UC Davis in 2021 using an RCBD with 10 replications and 3-row small plots (1.86 m$^2$) as experimental units. We measured 4 spikes per plot (subsamples) and harvested the small plots with a combine.

## Gene expression studies

To test if the differences in SNS between H3 and H1 were associated with different levels of *WAPO-A1* expression, we compared the transcript levels of this gene in developing spikes of homozygous H1 and H3 sister lines derived from HIF #120. *WAPO-A1* transcript levels were determined by qRT-PCR using A-genome-specific primers WAPO-A1-RT-F2 and WAPO1-A1-RT-R2 and PCR conditions described in our previous study [10]. Reactions were performed on an ABI 7500 Fast Real-Time PCR System (Applied Biosystems) using Fast SYBR GREEN Master Mix. Transcript levels were expressed as fold-*ACTIN* levels using the 2$^{\Delta CT}$ method.

Plants were grown in 1.4 L cones placed in CONVIRON growth chambers under 16 h light at 22˚C (330 mol intensity) and 8 h darkness at 17˚C for 25–30 days. For each replicate, we pooled 3–8 developing spikes from the main tiller when the lemma primordia were visible (W3.25) in the first and second experiments or when the floret primordia were present (W3.5) in the third experiment. Spike developmental stages are based on the Waddington scale [41]. We analyzed the first two experiments at W3.25 together using experiment as block, and the third experiment at W3.5 separately.

To compare the transcript levels of class-B, -C and -E MADS-box genes in Kronos and the *wapo1* mutant, we extracted RNA from developing spikes at the stamen primordia stage (~W4.0). A total of 12 developing spikes were pooled per replicate. Plants were grown in one-gallon pots in a CONVIRON growth chamber under similar conditions as described above. RNA extraction and expression analyses were done as described previously [42]. Transcript levels of *WAPO-A1* in the transgenic plants were conducted in a similar way, but developing spikes were harvested at stage W3.5 and 5–6 developing spikes were pooled per replicate.

For each transgenic construct, we selected one sister line without the transgene (WT), one with weak spike phenotype (e.g. increased SNS) and one with strong spike phenotypes (naked pistils and small terminal spikelet) and analyzed *WAPO-A1* transcript levels using primers *WAPO-A1*-RT-F2 and *WAPO-A1*-RT-R2 (S1 Table). We grew 24–36 T$_2$ progeny from each of the selected T$_1$ plants in a growth chamber and extracted RNA from pools of 6–9 developing spikes at the floret primordia stage (W3.5).

## *In situ* hybridization

We performed *in situ* RNA hybridization following the protocol described previously [21]. Tissues were obtained from developing spikes of diploid *T. monococcum* (accession PI 167615) and tetraploid wheat cultivar Kronos. cDNAs obtained from *T. monococcum* and Kronos were used to amplify *WAPO1* genes for the *in vitro* transcription reaction. We designed wheat A-genome specific primers appended with promoter sequences of T3 and T7 (S1 Table). Probes were synthesized using T3 (sense probe) or T7 (antisense probe) RNA Polymerase (Roche) and labelled with Digoxigenin-11-UTP (Roche). The forward primer P3-WM-APO1-T3-F$_{1400}$ starts 57 bp upstream of the stop codon and two alternative reverse primers P4-WM-A-PO1-T7-R$_{1649}$ and P5-WM-APO1-T7-R$_{1843}$ end in the 3' UTR and include a total of 266 and 458 bp respectively (S1 Table). The P3-P4 and P3-P5 probes showed the same specificity. The color reaction was stopped at 48 or 72 hours, and images were taken by Nikon Ti Microscope equipped with a DS-Fi2-U3 camera.

## Statistical analyses

The interaction between *WAPO-A1* and *WAPO-B1* was tested using a 2 x 2 factorial ANOVA using homeologs as factors and alleles (WT and CRISPR mutants) as levels. The four possible homozygous classes–WT, *wapo-A1*, *wapo-B1*, and *wapo1* double mutant–were selected from the $F_2$ progeny of an $F_1$ plant from the cross *wapo1* x Kronos WT without the CRISPR-Cas9 transgene.

For the transgenic lines TmDV92, LDN-C47, and LDN-F47, we generated three to five different transgenic events. Since the number of $T_1$ non-transgenic sister lines was small for each event, we pooled the non-transgenic sister lines from the different events generated with the same construct. We then compared the different transgenic events with the pooled non-transgenic lines with the same construct using Dunnett tests. All statistical analyses were performed with SAS version 9.4. Homogeneity of variance was tested using the Levene's test and normality of residual using the Shapiro-Wilks test as implemented in SAS v9.4. If necessary, data was transformed to restore the assumptions of the ANOVA. Data and descriptive statistics are provided in S1 Data file in Excel format, with data for the same figures and/or supplemental tables organized within a spreadsheet named with letters A to H.

## Supporting information

**S1 Table. Primers used in this study.**
(DOCX)

**S2 Table. $F_2$ CRISPR loss-of-function mutants *WAPO-A1* x *WAPO-B1* factorial ANOVA for spikelet number per spike (SNS), heading date and leaf number of the main tiller at heading.**
(DOCX)

**S3 Table. ANOVAs for 2020 and 2021 experiments testing the effect of H3 and H1 haplotypes on SNS, grain number per spike (GNS), grain yield, thousand kernel weight, and heading date.**
(DOCX)

**S4 Table. ANOVAs for 2021 experiment to evaluate the effect of *WAPO-A1* haplotype H2 introgressed into tetraploid Kronos and hexaploid high-biomass line GID4314513.**
(DOCX)

**S1 Fig. Expression analysis of *WAPO-A1* and *WAPO-B1* homeologs in developing spikes.**
(DOCX)

**S2 Fig. Strongest floral abnormalities in Kronos plants transformed with *WAPO-A1* genomic regions of LDN-F47-2.**
(DOCX)

**S3 Fig. *In situ* hybridization of *WAPO1* probe amplified from *T. monococcum* in developing spikes of diploid wheat *T. monococcum*.**
(DOCX)

**S1 Data. Data A.** Supporting data for Fig 1A–1C. Spikelet number per spike (SNS) in lines derived from Kronos EMS loss-of-function mutant K4222 for *wapo-A1*. **Data B**. Supporting data for Fig 2. Spikelet number per spike (SNS) and days to heading (DTH) in transgenic $T_1$ plants (different events) relative to pooled non-transgenic plants. **Data C**. Supporting data for Fig 3A–3D. Frequency of floral abnormalities in 14 Kronos plants with loss-of-function CRISPR mutations in both *wapo-A1* and *wapo-B1* (32 spikelets, 91 florets). **Data D**.

Supporting data for Fig 3E–3G. Expression of MADS-box genes involved in floral development in WT Kronos and *wapo1*-null (*wapo-A1 wapo-B1*) CRISPR truncation mutants. **Data E**. Supporting data for Fig 5. *WAPO-A1* transcript levels and fertility in Kronos transgenic plants transformed with *WAPO-A1* driven by its natural promoter. **Data F**. Supporting data for Fig 7A–7D and S3 Table. Split plot randomized complete block design field experiment testing the effect of *WAPO-A1* H1 and H3 haplotypes. **Data G**. Supporting Data for Fig 7E and 7F. Expression of *WAPO-A1* in homozygous H1 and H3 sister lines derived from HIF #120. **Data H**. Supporting data for Fig 8 and S4 Table. Effect of *WAPO-A1* haplotype H2 introgressed into tetraploid Kronos (CRD, 9 reps.) and high-biomass hexaploid line GID4314513 (RCBD, 10 blocks) in field experiments performed in 2021.
(XLSX)

## Acknowledgments

We thank Mariana Padilla and Oswaldo Chicaiza for excellent technical assistance in designing, managing, and collecting all field experiment data and to Juan Debernardi for reviewing the manuscript and valuable suggestions. We also thank André Schönhofen, Xiaoqin Zhang, and Priscilla Glenn for their help with the introgression of the H2 haplotype from common wheat into Kronos.

## Author Contributions

**Conceptualization:** Jorge Dubcovsky.

**Data curation:** Jorge Dubcovsky.

**Formal analysis:** Saarah Kuzay, Huiqiong Lin, Daniel P. Woods, Jorge Dubcovsky.

**Funding acquisition:** Jorge Dubcovsky.

**Investigation:** Saarah Kuzay, Huiqiong Lin, Chengxia Li, Daniel P. Woods, Tianyu Lan, Maria von Korff.

**Methodology:** Saarah Kuzay, Huiqiong Lin, Chengxia Li, Daniel P. Woods, Jorge Dubcovsky.

**Project administration:** Jorge Dubcovsky.

**Resources:** Saarah Kuzay, Huiqiong Lin, Chengxia Li, Shisheng Chen, Junli Zhang.

**Supervision:** Chengxia Li, Daniel P. Woods, Maria von Korff, Jorge Dubcovsky.

**Visualization:** Saarah Kuzay, Huiqiong Lin, Chengxia Li, Daniel P. Woods, Jorge Dubcovsky.

**Writing – original draft:** Saarah Kuzay.

**Writing – review & editing:** Saarah Kuzay, Huiqiong Lin, Chengxia Li, Shisheng Chen, Daniel P. Woods, Junli Zhang, Tianyu Lan, Maria von Korff, Jorge Dubcovsky.

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
