## [Decision Letter · Decision Letter 0]

26 Aug 2021

Dear Dr Dubcovsky,

Thank you very much for submitting your Research Article entitled 'WAPO-A1 is the causal gene of the 7AL QTL for spikelet number per spike in wheat' to PLOS Genetics.

The manuscript was fully evaluated at the editorial level and by two independent peer reviewers. The reviewers appreciated the attention to an important problem, but raised some substantial concerns about the current manuscript. Based on the reviews, we will not be able to accept this version of the manuscript, but we would be willing to review a much-revised version. We cannot, of course, promise publication at that time.

If you decide to revise the manuscript for further consideration at PLOS Genetics, please aim to resubmit within the next 60 days, unless it will take extra time to address the concerns of the reviewers, in which case we would appreciate an expected resubmission date by email to plosgenetics@plos.org.

[LINK]

We are sorry that we cannot be more positive about your manuscript at this stage. Please do not hesitate to contact us if you have any concerns or questions.

Yours sincerely,

Sarah Hake

Associate Editor

PLOS Genetics

Li-Jia Qu

Section Editor: Plant Genetics

PLOS Genetics

oth reviewers and myself appreciate the efforts of this work and the thorough genetics involved. I would have liked SEMs or in situs but realize that they may or may not be informative.

Reviewer 1 brought up many important comments that need to be addressed in order to be considered for publication in Plos Genetics. Please fix the areas of the manuscript and respond to all those comments plus my few ones below. I agree that it would be good to know if the MYC tagged protein is stable. MYC antibodies are available and this should not be a difficult experiment. Be sure to use inflorescences of the appropriate stage.

Here are a few additional minor comments that can be addressed.

I believe Kronos has the H1 haplotype from Table 1. perhaps make that obvious at first mention of Kronos.

line 208-. Mention is made of transcript levels for one line. Are the data for all the lines in the supplemental figure? Is this worth showing in the regular figures?

The figures of the floral organs needs a wild-type panel. Do these abnormalities interfer with fertility? I know the SNS number increases but are the florets all fertile?

line 275.observed in some plants. Please give frequency

line 320. end of sentence is missing

please be consistent with colors in the figures (I think this was mentioned also by reviewer 1).

line 490. trade off that (not trade off hat).

Reviewer's Responses to Questions

**Comments to the Authors:**

Reviewer #1: In this manuscript, Kuzay and Lin et al. investigate the genetic regulation of spikelet number in wheat by continuing their analysis of a quantitative trait locus on chromosome 7AL, for which they had previously proposed an underlying candidate gene, WAPO1. WAPO1 is an orthologue of ABERRANT PANICLE ORGANIZATION1 from rice – while their previous work supported their conclusion, the co-submission of two articles about the QTL meant that rigorous genetic validation had not been performed to confirm WAPO as the candidate, thus underlying the basis of this paper. Here, the authors provide further genetic proof of the role for WAPO in controlling spikelet number and floral architecture traits in wheat through analysis of TILLING mutant lines, transgenic lines that express the different allelic variants of WAPO and overexpress WAPO, as well as lines where WAPO has been edited using CRISPR. On the whole, the work provides solid genetic proof that the gene controls spikelet number in wheat. I do however have a number of concerns that should be addressed by the authors.

MAJOR COMMENTS

1)There are examples throughout the manuscript where the authors have presented statements about quantitative differences between genotypes but have not supported them with data in the figures or supplementary information. Can the authors please provide graphs or tables support this information because it is difficult to assess variation or the replication of this data. Also, it is not clear why the presented percentages do not include an indication of variation in the data (for example, mean � SEM). This comment also relates to examples where the authors have declared a correlation between traits, but not shown the phenotypic scores to support the associations.

2)In relation to the lines that overexpress the MYC-tagged version of WAPO, do the authors have any data to suggest that a functional protein is being produced in these lines? It is inconsistent with the transgenic lines that express LDN-F47 and LDN-C47 (Figure 2) that the UBI lines do not show a significant difference in spikelet number, and it indicates that the proteins may not be functional. Also, while I appreciate that they show higher expression in leaves, this does not necessarily mean the gene is overexpressed in the developing inflorescences. Is there any data showing higher expression in the developing inflorescences? Given the statement on L315, the authors have obviously considered the potential of the MYC protein interfering with the function of the protein, but they have not tested activity. I suggest the authors consider whether they need to include the data on the UBI overexpressing lines – does it add anything given the results of the LDN/TmDV92 transgenic lines, which are independent events?

3)Following on from this point, I am concerned about the comparison between transgenic lines on L322-326, and whether the phenotypes really are similar enough to conclude that the UBI and the genomic constructs both express active proteins. Firstly, it’s not clear based on the text used here whether the authors are trying to say that there were no significant phenotypic differences within plants of the same transgenic event, or among plants from the different transgenic events, or both? In the current version, I read it to mean that there are no significant phenotypic differences within plants of different transgenic events – if so, then I do not agree with their conclusion. For example, they do not share an effect on spikelet number, and the differences in heading date are difficult to examine because there is a 20-day difference in flowering time between the controls used for the LDN/TmDV92 transgenic lines and the UBI transgenic lines – why is this so, are they grown under different conditions/photoperiods? I agree that there are similar differences in floral architecture traits. I would like to ask whether the authors have considered the effect of the differences in higher expression of these two sets of transgenic lines – for example, if I interpret the graphs correctly, the UBI transgenic lines express WAPO >100-fold relative to wild-type, while the genomic constructs express the gene 8-10 fold higher (??) What affect would this difference have on the phenotypes? Again, I would like to ask the authors if they consider the genomic transgenic lines to be more insightful than the UBI transgenic lines, and whether the UBI transgenic lines should be included in the manuscript?

4)L286-288 – the authors should be careful with their conclusion about the association of more spikelets and the production of smaller distal spikelets in the WAPO1 transgenic lines. This is because the determination of spikelet number is temporally separate from the decision of a spikelet to form fertile florets, and without analysis of the developing inflorescences from the double ridge stage until emergence of the inflorescence, this conclusion may be too strong. For example, the spikelets of the transgenic lines may not be any different from the control lines during early stages, and the fertility of florets within the distal spikelets would be determined much later. There are also other reasons why the florets within distal spikelets could be infertile, including the floral architecture phenotypes shown be the authors in these lines – together, I recommend the authors reconsider this conclusion/statement, or ask that they perform further experiments to support it.

5)L368-371 – In my opinion, the statistical analysis performed across the two developmental stages is unnecessary. It is perfectly reasonable to expect that WAPO could show more difference in expression at one particular stage (e.g. W3.5), but not all (W3.25), and the separation of the two stages tells us something important about the gene function that is lost by attempting to combine the stages – for example, this result suggests that the gene is being suppressed in H1 lines as inflorescence development progresses, but not in the H3 lines. Potentially the effect of the haplotypes occurs during later stages.

6)L462 – it is not true that paired spikelets form more frequently at the base of the spike: the Ppd-1 publication shows that they form predominantly within the central region.

7)L506 – in relation to the conclusion about not “significantly affecting heading time”, the authors should clarify this statement by including “under field conditions”, as experiments with the transgenic lines showed delayed heading time under glasshouse conditions.

MINOR COMMENTS

1)L107 - Can the authors check the names of the ancestral alleles and correct the names if they agree that it is not necessary to include “1” after Wapo, here.

2)L131 – can the authors include detail in this sentence about how many amino acids are in the WAPO1 protein to support the statement about the mutant protein being truncated by 51%?

3)L145 – the use of the term “demonstrate” seems too strong here given the authors only examined one mutant here that could be influenced by associated background mutations. I understand the further work to come in the ms can support use of the term “demonstrate”, but here I recommend they use “indicate” or “suggest”.

4)For Figure 2, how many spikes were scored to generate this data? Can the authors include detail in the figure legend.

5)As a general comment, the order of data should be corrected in the graphs to have the wild-type genotype on the left and the mutant or transgenic lines on the right. For example, in Figure 1A-C, WT should be on the left followed by wapo-A1. Also consider this idea for Figures 5 and 6 in reference to the haplotypes.

6)Why are images within Figures 3 and 4 cut off in the figure panels? Are they of organs that do not show traits?

7)For Figure 5E-F, I recommend the authors describe the difference in expression in terms of “X-fold” rather than percentage difference.

8)For Figure 5, I found it confusing that the x-axis labels include Kronos and Rusty – I recommend the authors use H1 and H3 with the designated names of the alleles in brackets (also for Figure 6 for H1 and H2). The use of the alleles would better reflect the HIFs that were used to test the effect of theses haplotypes/alleles – in its current state, the names indicate that it the analyses were performed using Kronos and Rusty rather than the more informative germplasm that the authors generated.

Reviewer #2: Following a previous study by Kuzay et al. (2019; Identification of a candidate gene for a QTL for spikelet number per spike on wheat chromosome arm 7AL by high resolution genetic mapping. Theor Appl Genet. 132:2689–705. doi: 10.1007/s00122-019-03382-5) authors provided a solid data set with implications for wheat inflorescence architecture. In the present manuscript, they convincingly showed that a specific allele of an F-box gene underlying the WAPO-A1 locus is indeed causative for the increase in spikelet number per spike. Using KO mutants of the A and B copies of the gene in tetraploid wheat in combination with transgenesis by overexpressing WAPO1 authors revealed that expression level differences are related to the reduced or increased spikelet numbers, respectively. Similar to previous studies in Arabidopsis and rice, also these authors found floral aberrations of mutants and OE-lines, suggesting that WAPO1 acts upon homeotic floral regulators, such B- and C-class TFs, which was further exemplified in transcript levels from their qRT-PCR tests. Authors concluded justifiably that WAPO1 most likely functions very similar to its previously shown roles in rice and At.

Albeit majorly confirmatory (in relation to previous works in At and rice!), this study is highly appreciated for wheat researchers and breeders, since it showcases what’s required for functional gene analyses. I only have a few points for improvement:

1) The literature for the WAPO1 gene has not been properly covered!! The gene was initially discovered independently by three groups in parallel. Therefore, the paper by Muqaddasi et al. (2019) [TaAPO-A1, an ortholog of rice ABERRANT PANICLE ORGANIZATION 1, is associated with total spikelet number per spike in elite European hexaploid winter wheat (Triticum aestivum L.) varieties. Scientific Reports 9:13853; doi.org/10.1038/s41598-019-50331-9] must be added! Please include in line 80, line 92 and for the discovery of the F-box domain substitution C47F.

2) The claim in line 71 that grain yield has low heritability: I think this general statement is simply not true. GY has often heritabilities around 0.5, which is relatively high for a complex trait. Heritabilities for GY are often low in low-yielding ENVs with strong GxE effects. But in high-yielding ENVs it is often the opposite. I therefore suggest the authors to rather say: “Identifying genes controlling total GY is challenging due to its complex quantitative nature and GxE interactions.” Please replace and update the citations #3 and #4.

3) The correct plural form of ‘palea’ is ‘paleae’; please change accordingly

**Have all data underlying the figures and results presented in the manuscript been provided?**

Reviewer #1: **No: **There are examples throughout the manuscript (e.g. Figure 3A-D, Figure 4) where the authors have provided statements about percentage differences between genotypes that have not been supported by quantitative data within the figure panels, or the supplementary figures. The authors should present quantitative data to support these statements.

Reviewer #2: Yes

PLOS authors have the option to publish the peer review history of their article (what does this mean?). If published, this will include your full peer review and any attached files.

Reviewer #1: No

Reviewer #2: No

---

## [Decision Letter · Decision Letter 1]

18 Dec 2021

Dear Dr Dubcovsky,

We are pleased to inform you that your manuscript entitled "WAPO-A1 is the causal gene of the 7AL QTL for spikelet number per spike in wheat" has been editorially accepted for publication in PLOS Genetics. Congratulations!

Yours sincerely,

Sarah Hake

Associate Editor

PLOS Genetics

Li-Jia Qu

Section Editor: Plant Genetics

PLOS Genetics

Comments from the reviewers (if applicable):

Reviewer's Responses to Questions

**Comments to the Authors:**

Reviewer #1: I thank the authors for their efforts to address my concerns and amend the manuscript accordingly. With the changes to the manuscript, and the inclusion of new data, I believe this will be an important piece of research for the wheat breeding and plant science communities.

Reviewer #2: Thanks very much for considering all of my previous points.

**Have all data underlying the figures and results presented in the manuscript been provided?**

Reviewer #1: Yes

Reviewer #2: None

PLOS authors have the option to publish the peer review history of their article (what does this mean?). If published, this will include your full peer review and any attached files.

Reviewer #1: **Yes: **Scott Boden

Reviewer #2: No

**Data Deposition**

http://datadryad.org/submit?journalID=pgenetics&manu=PGENETICS-D-21-00999R1

**Press Queries**

---

## [Editor Report · Acceptance letter]

10 Jan 2022

PGENETICS-D-21-00999R1 

WAPO-A1 is the causal gene of the 7AL QTL for spikelet number per spike in wheat 

Dear Dr Dubcovsky, 

We are pleased to inform you that your manuscript entitled "WAPO-A1 is the causal gene of the 7AL QTL for spikelet number per spike in wheat" has been formally accepted for publication in PLOS Genetics! Your manuscript is now with our production department and you will be notified of the publication date in due course.

With kind regards,

Livia Horvath

PLOS Genetics

On behalf of:
